# Zn and Ag Doping on Hydroxyapatite: Influence on the Adhesion Strength of High-Molecular Polymer Polycaprolactone

**DOI:** 10.3390/molecules27061928

**Published:** 2022-03-16

**Authors:** Jiaming Song, Xuehan Li, Naiyu Cui, Xinyue Lu, Jiahao Yun, Qixuan Huang, Yunhan Sun, Eui-Seok Lee, Hengbo Jiang

**Affiliations:** 1The Conversationalist Club, School of Stomatology, Shandong First Medical University & Shandong Academy of Medical Sciences, Tai’an 271016, China; jiamingsongpp@outlook.com (J.S.); xuehanlii@outlook.com (X.L.); cuinaiyuuu@outlook.com (N.C.); xinyuelu111@outlook.com (X.L.); yun24020376@outlook.com (J.Y.); xianliang_h@163.com (Q.H.); docyunhan@outlook.com (Y.S.); 2Department of Oral and Maxillofacial Surgery, Graduate School of Clinical Dentistry, Korea University, Seoul 08308, Korea

**Keywords:** hydroxyapatite/polycaprolactone, ionic doping, adhesion strength, density functional theory

## Abstract

In this study, density functional theory was employed to calculate the adsorption of polycaprolactone (PCL) by pure hydroxyapatite (HA), Zn-doped HA, and Ag-doped HA, and the interaction of PCL on the surface of HA (001) was simulated. The results show that there was significant electron transfer between the carbonyl O in PCL and the Zn, Ag, and Ca in HA, forming coordinate bonds. The binding energies of Ag-doped HA/PCL and Zn-doped HA/PCL were much higher than those of HA/PCL. HA doped with Ag had the highest binding energy to PCL. Therefore, we believe that when HA is doped with Ag atoms, its adsorption capacity for PCL can be increased. The results obtained in this study can be used as a guide for the development of HA/PCL bone graft composite material doped with appropriate metal ions to improve its adsorption capacity.

## 1. Introduction

With an increase in the average human age, the prevalence of bone diseases has increased in the population worldwide [1]. Therefore, it is necessary to develop biomaterials that can provide osteogenicity [2]. Traditional bone graft materials, such as autologous bone and allogeneic bone grafts, require the surgical removal of bones, and there is a risk of disease transmission in the case of an allogeneic bone graft [3]. As an inorganic substance that is similar to the human bone, hydroxyapatite (HA) has good biological activity and osteoconductivity and can directly participate in the cell division and mineralization of bones [4]. It is considered an effective material for bone-tissue regeneration [5]. However, due to its poor mechanical properties (such as high brittleness and low strength) [6], the load-bearing applications become challenging [7]. Polycaprolactone (PCL) is a biodegradable polymer that has good biocompatibility [8]. Compared with HA materials, composite materials that are composed of PCL and HA can considerably compensate for this defect and effectively improve their mechanical properties [9]. It can support the proliferation of bone-marrow-derived mesenchymal stem cells and dental pulp stem cells and has a good supporting role in terms of bone defects. Therefore, composite materials composed of PCL and HA are often used in research on bone regeneration [10,11].

Interface bonding plays an important role in the strengthening and toughening of composite material systems. Due to the significant chemical differences between HA and polymers, the composite material exhibits poor interfacial adhesion, resulting in a decrease in mechanical strength [12,13]. This shortcoming can affect the mechanical properties of the composite materials; therefore, improving the adhesion of PCL and HA is of great significance, to enhance the physical and mechanical properties of PCL/HA composite materials. Previous studies have mainly focused on adjusting the binding ability of PCL and HA by varying the preparation conditions. For example, HA powder has been prefabricated in PCL by sol–gel method to obtain a uniform PCL/HA composite material and avoid HA agglomeration in the PCL matrix [13]. In addition, the HA particle size also affects the interfacial bonding ability of PCL. An inferior interface between micron-sized HA and PCL shows many cracks and voids. Good interfacial adhesion was observed between the HA nanoparticles and the substrate [14].

An isomorphic replacement of multiple ions, such as Zn, Ag, Mg, and Sr ions, can be performed in HA. Substitutions affect the stability [15,16,17], antibacterial properties [18,19], biological activity [20,21,22,23], and bonding properties [24] of HA. Sun et al. found that doping with Zn and Ag atoms significantly improved the bonding strength of HA and Ti [25]. Therefore, this research proposes a new method of improving the adhesion between PCL and HA by doping with Zn or Ag atoms. With the development of density functional theory and computer performance, the accuracy of DFT calculation has been greatly improved, because DFT calculation can intuitively show the interaction mechanism between molecules. It plays an important role in materials science. At present, computational simulation methods can not only study existing materials, but also predict new materials. The purpose of this study was to investigate the mechanism of adsorption of PCL, occurring via the functional groups in PCL, onto pure HA, Zn-doped HA, and Ag-doped HA based on density functional theory (DFT), to provide a theoretical basis for improving the adhesion strength between PCL and HA.

## 2. Theoretical Methods

### 2.1. Model Building

HA has a hexagonal structure with a point group in the P63/m space. In our previous study, we replaced Ca(II) with Zn or Ag atoms to create Zn-doped HA and Ag-doped HA models. The (001) crystal plane of HA was selected for further study. We optimized the obtained unit cell from the three models. A 30 Å vacuum layer was established to avoid interactions between the upper- and lower-unit cells. The optimized unit cell was then expanded into a 2 × 1 × 1 supercell.

Since our research focuses on the mechanism of action between the functional groups in PCL and HA, we refer to the method of studying the polymer and crystal surface adsorption by Ian Streeter et al. [26]. We selected a segment of the PCL polymer for examination, which included all the functional groups of PCL: a carbonyl group, an ether bond, and an alkane chain (Figure 1).

### 2.2. DFT Calculation

We optimized the geometry of all the adsorption structures. CP2K software was used for calculations [27]. A DZVP-MOLOPT-SR-GTH basis set and Perdew–Burke–Ernzerhof functional [28] were applied to exchange related functionals and 500Ry plane wave cutoff. A DFT-D3 dispersion correction was added [29] because of weak interactions in the system.

## 3. Results and Discussion

### 3.1. Lattice Parameters of Pure HA, Zn-Doped HA, and Ag-Doped HA

The lattice parameters for pure HA, Zn-doped HA, and Ag-doped HA, listed in Table 1, were obtained by first-principle calculations. The calculated lattice parameters of HA were slightly higher than the experimental lattice parameters [30] because, during optimization, the unit cell is slightly enlarged. The lattice parameters, a and b, of Zn-doped HA are 1.867% and 0.121% lower than those of pure HA, respectively, and c is increased by 0.2%. The lattice parameter, a, of Ag-doped HA decreased by 0.888%, while b and c increased by 0.179% and 0.292%, respectively. Because of the defects of HAp lattice structure caused by the doping of Zn and Ag, the cell parameters of doped hydroxyapatite decreased.

### 3.2. Electrostatic Potential (ESP)

The ESP is commonly used to express th e electron density on the surface of a molecule. By analyzing the distribution of the molecular ESP, the mode and mechanism of the intermolecular interactions can be predicted and explained [31,32]. Figure 2 shows the possible active sites of HA and PCL during the adsorption process. The blue area represents a positive ESP and shows electrophilicity, while the red area represents a negative ESP and shows nucleophilicity.

For pure HA, it was found that the positive ESP is mainly distributed near the Ca atom, and the negative ESP is mainly distributed near the phosphoric acid group, which shows that the Ca atom is an electrophilic site, while the phosphate group is a nucleophilic site. For Zn-doped HA and Ag-doped HA, the positive ESP is mainly distributed near the Ca, Zn, and Ag atoms, while the negative ESP is mainly distributed near the phosphoric acid group.

As shown in Figure 3, there are two areas of negative ESP, O1 and O2, on the surface of PCL, which can be bound to the positive Ca atoms on the surface of HA. Of these, O1 has the highest negative ESP. Therefore, theoretically, when PCL interacts with the pure and doped HAs, O1 in PCL can easily form coordinate bonds with Ca, Zn, and Ag atoms in HA.

### 3.3. Adsorption Energy and Geometry

The adsorption strength between the two compounds was evaluated by measuring the adsorption energy between HA and PCL. Table 2 shows the adsorption energy of PCL on the surfaces of pure HA, Zn-doped HA, and Ag-doped HA. The adsorption energy was calculated as follows:$$E_{ads}=\left(E_{\mathrm{SUB}}+E_{\mathrm{PCL}}\right)-E_{\mathrm{SUB}+\mathrm{PCL}}$$

Here, $E_{\mathrm{SUB}+\mathrm{PCL}}$, $E_{\mathrm{SUB},}$ and $E_{\mathrm{PCL}}$ represent the energy of the composite structure after the structure optimization, the substrate, and the PCL, respectively.

**Table 2 molecules-27-01928-t002:** Adsorption energy of PCL and HA, Zn-doped HA, and Ag-doped HA (Units: eV).

Model	Adsorption Energy
HA-PCL	1.255
Zn-doped HA-PCL	1.315
Ag-doped HA-PCL	1.418

The adsorption energies are all positive; therefore, the adsorption between PCL and HA is thermodynamically favored. More importantly, the adsorption energy of PCL on Zn- and Ag-doped HA is greater than the adsorption energy of PCL on pure HA. This shows that the doping of Zn and Ag on HA improves adsorption with PCL. Moreover, the adsorption energy of PCL on the surface of HA doped with Ag atoms increased by 13.0%, while the adsorption energy of HA with doped Zn atoms increased only by 4.8%. The results show that the binding of PCL can be enhanced by doping HA with Ag atoms.

Figure 4 shows the adsorption structure of PCL on different HA surfaces. For pure HA, the carbonyl O atom in PCL forms a coordinate bond with the Ca atom, with a bond length of 2.328 Å. When PCL is adsorbed, the distance between Ca and the surrounding O atoms changes. For HA doped with Zn and Ag atoms, the carbonyl O atoms in PCL will also form coordinate bonds with Zn and Ag atoms, with bond lengths of 2.146 Å and 2.309 Å, respectively. In addition, Zn and Ag atoms move upward and close to the carbonyl O atoms, which facilitates the adsorption of PCL. The other parts of PCL have similar adsorption geometries on the surfaces of pure HA, Zn-doped HA, and Ag-doped HA, indicating that the strength of the coordinate bond is the main factor that affects the strength of the adsorption of PCL and HA.

### 3.4. Charge-Density Difference Analysis

The adsorption between PCL and HA leads to a redistribution of the electrons. Analysis of the charge-density difference therefore helps in studying the adsorption between PCL and HA. The formula for charge-density difference calculation is as follows:$$\Delta \rho =\rho_{\mathrm{SUB}+\mathrm{PCl}}-\rho_{\mathrm{SUB}}-\rho_{\mathrm{PCL}}$$

Here, $\rho_{\mathrm{SUB}+\mathrm{PCl}}$ represents the charge density of the composite structure, $\rho_{\mathrm{SUB}}$ represents the charge density of the substrate after PCL removal, and $\rho_{\mathrm{PCL}}$ represents the PCL charge density.

Figure 5 shows the charge-density difference, where the yellow area indicates a depletion of electrons and the cyan area indicates an accumulation of electrons. It can be seen that Ca, Zn, and Ag atoms accumulate electrons, indicating that they act as electron acceptors (Lewis acids). The depletion of electrons around the carbonyl O in PCL indicates that it acts as an electron donor (Lewis base), which corresponds to the result of our ESP analysis. Moreover, the carbonyl O atom in PCL has an obvious electron transfer with Ca, Zn, and Ag atoms, respectively, which shows the formation of a coordinate bond between them.

### 3.5. Molecular Orbital

Figure 6 shows the highest occupied molecular orbital (HOMO) and lowest unoccupied molecular orbital (LUMO) of PCL. We found that the HOMO is significantly distributed on the carbonyl O atom of PCL; hence, the carbonyl O atom on PCL has a strong nucleophilicity.

Among the HOMOs and LUMOs of pure HA, Zn-doped HA, and Ag-doped HA (Figure 7), we found that Ag-doped HA has more LUMOs on Ag atoms, which indicates a relatively strong electrophilicity of the Ag atoms. Ag-doped HA has the strongest bonding with the carbonyl O atom of PCL, which results in a maximum adsorption energy.

## 4. Conclusions

DFT was employed to study the mechanism of interactions of pure HA, Zn-doped HA, and Ag-doped HA with PCL. The analysis was performed on the surface of HA (001). The results show that the carbonyl O in PCL has significant electron transfer with the Ca, Zn, and Ag atoms on HA, which indicates the formation of coordinate bonds. The adsorption capacity of Zn-doped HA and Ag-doped HA with PCL can be significantly improved by doping HA with Ag and Zn atoms. The adsorption energy of Zn-doped HA and Ag-doped HA with PCL was much higher than that of pure PCL. Ag-doped HA had the highest adsorption energy with PCL. Therefore, doping with metal ions (Zn and Ag atoms) can enhance the adhesion strength of HA and PCL, and the adhesion strength to PCL can be significantly increased by replacing Ca(II) in HA with Ag atoms.

## Figures and Tables

**Figure 1 molecules-27-01928-f001:**
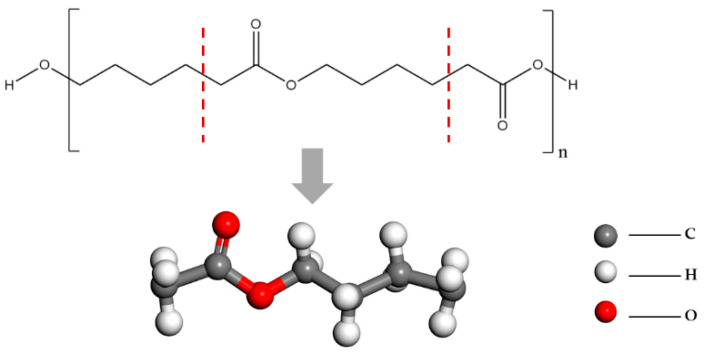
Structure of PCL and the fragments selected in this study.

**Figure 2 molecules-27-01928-f002:**
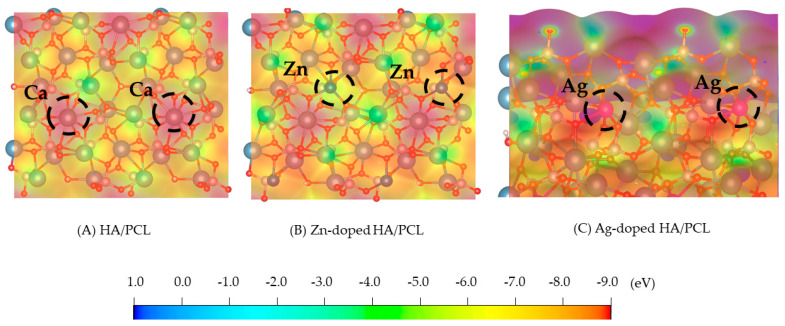
ESP of PCL on different surfaces of HA, Zn-doped HA, and Ag-doped HA.

**Figure 3 molecules-27-01928-f003:**
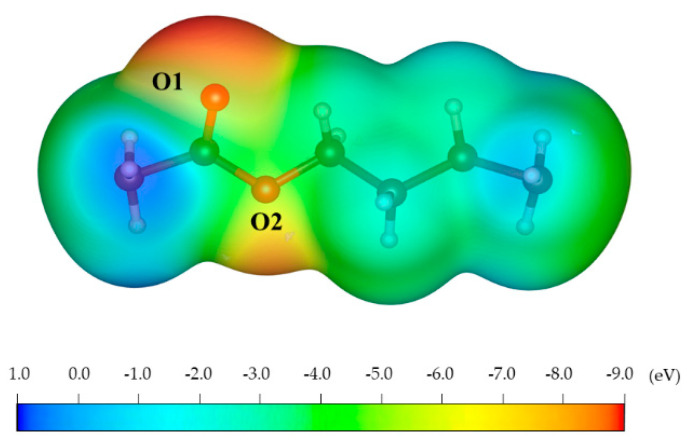
ESP of PCL.

**Figure 4 molecules-27-01928-f004:**
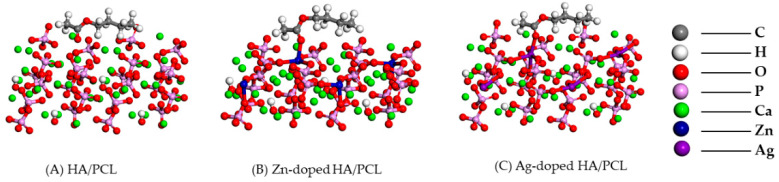
Adsorption structure of PCL on different surfaces of various HA structures.

**Figure 5 molecules-27-01928-f005:**
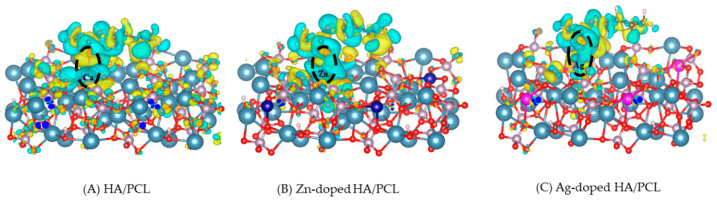
Charge-density difference of PCL on different HA surfaces.

**Figure 6 molecules-27-01928-f006:**
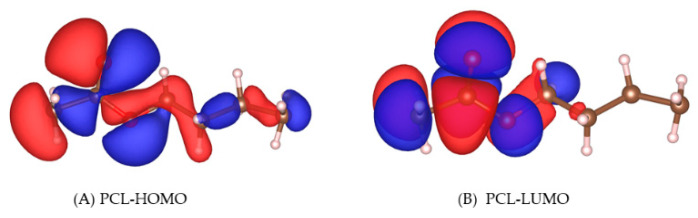
HOMOs and LUMOs of PCL.

**Figure 7 molecules-27-01928-f007:**
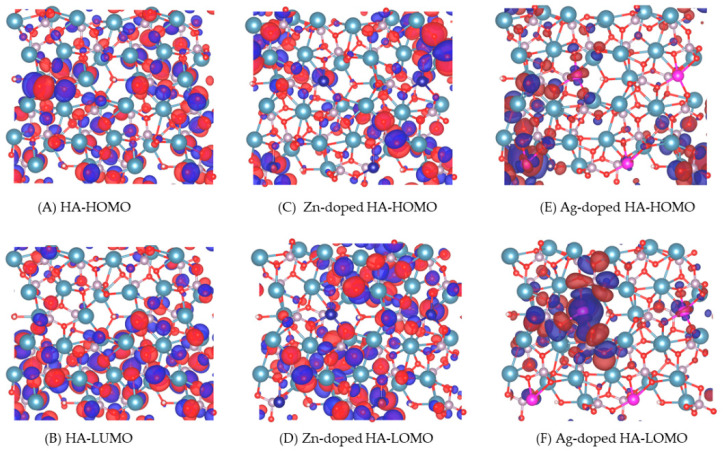
HOMOs and LUMOs of HA, Zn-doped HA, and Ag-doped HA.

**Table 1 molecules-27-01928-t001:** Lattice parameters of different HA models.

Model	a (Å)	b (Å)	c (Å)	α (°)	β (°)	γ (°)	V (Å^3^)
HA	9.569	9.569	6.984	90.000	90.000	120.00	553.828
Zn-doped HA	9.390	9.557	7.005	89.875	89.947	119.309	548.175
Ag-doped HA	9.484	9.586	7.005	90.349	89.616	119.660	553.385
Experimental HA [30]	9.551	9.551	6.843	—	—	—	540.59

## Data Availability

The data used to support the findings of this study are included within the article.

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
