# Peer review of "Zn and Ag Doping on Hydroxyapatite: Influence on the Adhesion Strength of High-Molecular Polymer Polycaprolactone"

_molecules, 2022, doi:10.3390/molecules27061928_

Round 1
Reviewer 1 Report
The article presents very interesting material on the emerging field of biotechnology. Describes in detail a new approach to improving
the adhesion between PCL and HA by doping with Zn or Ag atoms, what can be used for development of HA / PCL bone graft composite. It is written clearly. It contains a properly conducted discussion. The literature cited is current.
Author Response
Thanks for your kind comments.
Reviewer 2 Report
I have carefully read the manuscript entitled “Zn and Ag Doping on Hydroxyapatite: Influence on the Adhesion Strength of High-molecular Polymer Polycaprolactone” and analyzed its potential for publication in the MDPI journal Molecules (ISSN 1420-3049).
In my opinion, the manuscript is interesting and involves recent topics.
I suggest accepting it for publication after minor revision.
The only points I would like to rise in my review are:
- Introduction contains some basic information on the application of hydroxyapatite/polycaprolactone composite in bone regeneration. However, the background concerning usage of insilico methods in studies of the issue and related topics that could provide reader with better understanding of scientific soundness of presented data is missing.
- Additionally I would expect the expression of the aim of the study including univocally pointed novelty of the study.
Author Response
- Introduction contains some basic information on the application of hydroxyapatite/polycaprolactone composite in bone regeneration. However, the background concerning usage of insilico methods in studies of the issue and related topics that could provide reader with better understanding of scientific soundness of presented data is missing.
Response:Thanks for your suggestion, we added a background on computational simulation in introduction “With the development of density functional theory and computer performance, the accuracy of DFT calculation has been greatly improved, because DFT calculation can intuitively show the interaction mechanism between molecules. It plays an important role in materials science. At present, computational simulation methods can not only study existing materials, but also predict new materials.”
- Additionally I would expect the expression of the aim of the study including univocally pointed novelty of the study.
Response: We propose a method to improve the binding ability between hydroxyapatite and PCL by adding hydroxyapatite. “DFT method was used to simulate the binding strength and mechanism between hydroxyapatite and PCL containing silver and zinc. The results showed that doping with metal ions (Zn and Ag atoms) can enhance the adhesion strength of HA and PCL, and the adhesion strength to PCL can be significantly increased by replacing Ca(II) in HA with Ag atoms.”
Reviewer 3 Report
In the present manuscript, the authors examine the adsorption of polycaprolactone on pure and doped hydroxyapatite by means of DFT calculations. They found that Ag atoms increase the adsorption capacity. The results seem reasonable in my opinion. I believe that the manuscript after revision can be accepted for Molecules.
-
Consider expand the explanation of the method of studying the polymer and crystal surface adsorption to this particular case, and not refer to a reference of other system.
-
In section 3.1., provide an explanation for the reduction of the cell parameter in Zn and Ag doped hydroxyapatite.
-
The author argues that “carbonyl O in PCL has significant electron transfer with the Ca, Zn, and Ag atoms on HA” based on Charge density difference and Molecular orbital analysis. For this reader, this analysis must be complemented with some sort of atomic charge calculations on these atoms.
Author Response
- Consider expand the explanation of the method of studying the polymer and crystal surface adsorption to this particular case, and not refer to a reference of other system.
Response: Due to the limited computational resources, it is very difficult to accurately calculate the adsorption between complete polymer and HAp. So we cut a small section containing polymer, including all the functional groups in the polymer. This method not only greatly improves the calculation speed, but also can accurately analyze the adsorption between polymer and various hydroxyapatite.
- In section 3.1., provide an explanation for the reduction of the cell parameter in Zn and Ag doped hydroxyapatite.
Response:Because of the defects of HAp lattice structure caused by the doping of Zn and Ag, the cell parameters of doped hydroxyapatite decreased.
- The author argues that “carbonyl O in PCL has significant electron transfer with the Ca, Zn, and Ag atoms on HA” based on Charge density difference and Molecular orbital analysis. For this reader, this analysis must be complemented with some sort of atomic charge calculations on these atoms.
Response: Thank you very much for your suggestion. We have calculated the atomic charge by bader charge analysis, but the calculated charge gains and losses of carbonyl O in PCL and Ca, Zn and Ag atoms in HA cannot be explained as the formation of coordination bonds between these atoms, because it may be caused by the chemical bonds formed between these atoms and other atoms. Therefore, in order to eliminate the influence of chemical bonds between these atoms and other atoms, we use charge density difference analysis to subtract the charge density of adsorption complex to HAp and PCL before adsorption. Thus, the charge transfer between carbonyl O in PCL and Ca, Zn and Ag atoms in HA is caused by the formation of coordination bonds. Charge transfer between charge difference analysis can be observed very intuitively through the difference of color distribution.
Reviewer 4 Report
The manuscript "Zn and Ag Doping on Hydroxyapatite: Influence on the Adhesion Strength of High-molecular Polymer Polycaprolactone" doesn't provide any experimental results, but only theoretical modeling. Presented modeling results are not significant if not supported with experimental results to validate them. The manuscript cannot be suitable for publication in the journal "Molecules".
Author Response
Thank you very much for your advice.
In this study, we only conducted theoretical simulation, which proved the feasibility of adding Zn and Ag to hydroxyapatite to improve the binding strength of PCL, providing useful theoretical support for subsequent experiments.